# Machine Learning in Prediction of IgA Nephropathy Outcome: A Comparative Approach

**DOI:** 10.3390/jpm11040312

**Published:** 2021-04-17

**Authors:** Andrzej Konieczny, Jakub Stojanowski, Magdalena Krajewska, Mariusz Kusztal

**Affiliations:** Department of Nephrology and Transplantation Medicine, Wroclaw Medical University, 50-556 Wroclaw, Poland; jakub.stojanowski@student.umed.wroc.pl (J.S.); magdalena.krajewska@umed.wroc.pl (M.K.); mariusz.kusztal@umed.wroc.pl (M.K.)

**Keywords:** artificial intelligence, IgA nephropathy, proteinuria, end-stage renal disease, chronic kidney disease

## Abstract

We are overwhelmed by a deluge of data and, although its interpretation is challenging, fortunately, information technology comes to the rescue. One of the tools is artificial intelligence, allowing the identification of relationships between variables and their arbitrary classification. We focused on the assessment of both the remission of proteinuria and the deterioration of kidney function in patients with IgA nephropathy, comparing several methods of machine learning. It is of utmost importance to respond to subtle changes in kidney function, which will lead to a deceleration of the disease. This goal has been achieved by analyzing regression techniques, predicting the difference in serum creatinine concentration. We obtained the performance of the tested models which classified patients with high accuracy (Random Forest Classifier showed an accuracy of 0.8–1.0, Multi-Layer Perceptron an Area Under Curve of 0.8842–0.9035 and an accuracy of 0.7527–1.0) and regressors with a low estimation error (Decision Tree Regressor showed MAE 0.2059, RMSE 0.2645). We have demonstrated the impact of both model selection and input features on performance. Application of machine learning methods requires careful selection of models and assessed parameters. The computing power of modern computers allows searching for the models most effective in terms of accuracy.

## 1. Introduction

The age of electronics challenges humans in the field of big data analysis. Huge amounts of data cannot be analyzed by an individual, but instead by information technology, with a dynamically developing branch called artificial intelligence (AI) which comes to the rescue. AI is a broad umbrella term for computer techniques mimicking human intelligence, in particular machine learning (ML) methods, gaining “experience” and “knowledge” during supervised training on data [1]. Deep learning is an even narrower concept, reaching a higher level of abstraction, using convolutional networks. ML aims to detect unknown regularities, formulates a decision template, finds non-obvious relationships between features, and provides approaches supported by big data analysis. The trained model enables the prediction of the occurrence of the disease in the future and allows for the assessment of illness progression.

IgA nephropathy (IgAN), the most frequent primary glomerulonephritis, leading to end-stage renal disease (ESRD), is a very heterogeneous disease [2]. Its symptoms vary from solely erythrocyturia and benign course to rapidly progressive glomerulonephritis, leading eventually to ESRD, requiring renal replacement (RRT). Therefore, it is of utmost importance to follow up the patient and it is imperative to assess the direction in which his condition is developing [3].

An example of effective ML is Multi-Layer Perceptron (MLP), an artificial neural network where all nodes are connected to each other between the layers node [1,4,5]. The advantage of neural networks over linear analysis is the capability of finding nonlinear relationships. An artificial neural network with one layer works like a line analyzer, MLP with two layers (one hidden layer) defines convex regions, while an MLP with three layers (with two hidden layers), decides arbitrarily with a complexity which is limited by the number of nodes [6]. The use of MLP is an introduction to deep learning, used in analyzing data in which the proximity of elements is important, e.g., photos or images where the sequence of pixels directly affects the whole picture.

This work aims to develop a diagnostic support system, assessing the importance of input features in a large primary database, and selecting the most important models for effective prediction and testing design built on these subsets of input data. We particularly focused on these issues to support the decision-making process.

Based on the database of patients, with biopsy-confirmed IgAN, the task of the program was to find templates predicting the progression of the disease with the greatest accuracy, and assess the expansion of the disease, with the narrowest error. Supervised training of classifiers allowed us to anticipate the progression of IgAN, while methods of regression were used to quantify and predict the difference in serum creatinine concentration.

## 2. Materials and Methods

### 2.1. Study Cohort

Patients with biopsy-proven IgAN from the Department of Nephrology and Transplantation Medicine at the Wroclaw Medical University were screened from January 2010 to November 2019. The exclusion criteria were as follows: (1) age < 18 years old; (2) secondary IgA deposition; (3) Glomerular Filtration Rate (eGFR) < 15 mL/min per 1.73 m^2^; (4) coexistence of other renal diseases; (5) lack of follow-up (<12 months); and (5) less than 7 glomeruli in a renal tissue section. Eighty IgAN patients were enrolled and followed up for a median of 4 years. IgAN was diagnosed and defined in immunochemistry microscopy by the presence of predominant mesangial IgA deposits with or without other immunoglobulins. The eGFR was calculated using the Modification of Diet in Renal Disease (MDRD) study equation. The details of pathological data were reviewed according to the Oxford classification criteria for IgAN (MEST-C) [5]. MEST-C scores were reported as follows: M0/M1 was defined as the absence or presence, respectively, of 50% of glomeruli showing hypercellularity, E0/E1 was defined as the absence or presence, respectively, of endocapillary hypercellularity, S0/S1 was defined as the absence or presence, respectively, of segmental sclerosis or tuft adhesions, and T0/T1/T2 was defined as the degree of tubular atrophy or interstitial fibrosis (<25%, 26–50%, >50%, respectively). In addition, global and segmental glomerulosclerosis was calculated as the proportion of involved sclerotic glomeruli divided by the total number of glomeruli. Interstitial inflammatory lesions were assessed in terms of the presence or absence of such lesions. Interstitial fibrosis and tubular atrophy were evaluated as the percentage of the affected cortical area, namely as mild, moderate, or severe (<25%, 25–49%, >50% respectively).

The study was conducted in accordance with the guidelines of the Declaration of Helsinki.

### 2.2. Endpoints, Outcome, and Model Classification

In our work, we focused on the use of ML in the prediction of kidney function deterioration in the course of IgAN. The main task of the program was to allocate patients’ records into appropriate groups. The classifying model was prepared on the training data and its effectiveness was checked on the basis of the test data. Patients were arranged into 3 groups: those with remission of proteinuria, defined as urine protein to creatinine ratio (UPCR) below 0.2 g/g (1); those in whom remission did not occur (0); and those in whom renal function did not deteriorate (2), i.e., the first and subsequent creatinine levels remained within the normal range. The effectiveness of the models was measured using accuracy, and different training and testing data were verified for each method. We have determined the maximum, mean, and minimum accuracy for each set. The diversification of the program input allowed to test the stability of ML, depending on the selected algorithm and input data.

In the second part, we estimated the deterioration of renal function, as expressed by the difference in creatinine concentration, compared to the baseline.

### 2.3. Machine Learning Models

We developed a program that tested various types of regressors, including AdaBoost Regressor, Support-Vector Regression, and Decision Tree Regressor.

Using the Python programming language and the available scikit-learn library, a program, creating 5 machine-learning models, was developed [6].
Gaussian Naive Bayes ClassifierSupport Vector MachineRandom Forest ClassifierK-nearest Neighbor ClassifierMulti-Layer Perceptron, which is also an example of an artificial neural network (ANN)

Gaussian Naive Bayes (GNB) is a simple classifier derived from the Bayesian theorem, modified to create models, based on normally distributed (Gaussian) data [6]. The variables are assumed to be independent of each other, which is a certain limitation for application to real-world problems.

Support Vector Machines (SVM) are a group of supervised machine learning methods. The basic principle of SVM is to divide the data set into subsets assigned to classes. It computes a hyperplane allowing allocation of the data into appropriate classes. This method is effective for data with many variables, even if the number of variables is greater than the number of samples [6].

There are ML methods that use several classifiers, and the final solution is the averaged values of individual classifiers. Random Forest Classifier (RF) creates a set of decision trees that are perturbed to avoid overfitting of the training data [6,7]. Particular outputs are combined into final-class classification. Individual decision trees can be interpreted easily by simply visualizing the tree structure. The advantage is that respective decision trees intrinsically perform feature selection by choosing appropriate split points.

K-nearest neighbor classifier (KNN) is a group of supervised learning methods. It differs from other approaches in that it does not lead to conclusions but stores training data in a performance-efficient data structure [6]. The unsupervised k-nearest neighbor method can be used to fill in missing data in the database, e.g., single cells. It may be used as pre-processing in conjunction with an appropriate supervised learning method. Null cells in a dataset can lead to improper behavior of the predictive model. Lost data can be supplemented by assigning the most appropriate values, based on the remainder of the data.

An MLP is an example of a simple ANN, mimicking the function of the human brain, in particular cognition and inference [6]. An MLP definitely has fewer neurons than the human brain, yet it is able to “conclude” on the basis of numbers and classify new data on its own. Neurons in the perceptron appear as nodes in the graph. Each node has a set of inputs, assigned weights, and the so-called bias value, allowing the shift of the activation function to the left or right, which may be crucial for successful learning. The value calculated at the node is the weighted average of the inputs and weights plus bias. The output value is expressed as the so-called value. An MLP is formed by the interconnection of several layers. Hidden layers are designed to buffer input values, which translates into network complexity, but at the same time allows very efficient output. The MLPs used in our experiment has 2 hidden layers that allow arbitrary classifications, limited by the number of nodes in the neural network.

The features constructed from tabular data are often correlated, so a small subset of features is responsible for most of the predictive power. An image is a two-dimensional space where a point with given coordinates is assigned a color value. Each column in the table is an additional dimension in the space analyzed by the model. A row in the table represents a point in the multidimensional space. Each row has many variables, e.g., age, sex, UPCR, serum creatinine concentration, etc. Multidimensional spaces pose a challenge for the exploration of deep neural network, hence their lower efficiency may result. Unlike in data from images, audio, and language, there can be little variation in a column of a table. Deep learning networks have filters that are sensitive to data proximity.

Accuracy is defined as:(1)$$ACC=\frac{True\;positives+True\;negatives}{Total\;population}$$

The program randomly divided the data into a training set and a test set in 42 ways. For each of the data splits, the program checked 39 k-Nearest Neighbor Classifiers, 43 Gaussian Naive Bayes Classifiers, 9 Random Forest Classifiers, 2401 Multi-Layer Perceptron Classifiers, and 99 Support Vector Machines. The Multi-Layer Perceptron consisted of 2 hidden layers, each with up to 50 neurons. The test was run on several input data sets that contained different parameters. We checked how the models behave depending on the available data. Fewer input parameters were allowed to be used as a screening test in smaller canters. (Figure 1) The other classifiers, except for the Gaussian Naive Bayes classifier, had one start parameter, independently selected from the input data. Our test took into account different values of the startup parameter.

Due to the small size of input data, the L-BFGS solver was used, an optimizer alternative to the default Adam solver [8]. It achieves better performance and allows faster descent to the optimal solution which is important when learning on a small dataset.

The main clinical goal is to correctly classify the data. The time it takes to check several models from which to choose the best one in each case is short, so instead of one model, a voting system can be chosen based on several other models. The one-vs-rest classification works in a similar way, where for each classifier, a class is fitted against all the other classes. In our work, we focused on the correct assignment of classes to testing records. We chose accuracy as the key performance parameter due to the unbalanced class distribution in our database [9]. The activation function for the hidden layer is Relu, the rectified linear unit function, given by the formula f(x) = max(0, x), which returns a value of a non-negative value argument, or returns zero if the argument is negative [4,10].

## 3. Results

### 3.1. Characteristics of the Study Population

Clinical, histopathological, and laboratory data of the study cohort are displayed in Table 1. Our program was provided with these data and then independently assessed feature importance and selected subsets, the analysis of which is at the heart of our work. The test group was selected randomly and shuffled to test the performance of the model.

### 3.2. The Performance of Classifiers

Our program was able to select the data with the highest importance in the prediction. The ML methods reflect, to a large extent, the human decision-making process. Parameters selected by the random forest classifier, on the basis of Gini validity, made it possible to identify optimal sets of features and eliminated the variables interfering with the classifier [11,12]. Then the algorithm selected subsets containing the most significant parameters and performed further tests for them. In this way, we could observe the influence of the selection of input data on the performance of the models. The Random Forest Classifier was the most stable algorithm, depending on input features. At the same time, the model is human-readable because it can be presented in the form of a decision tree.

Figure 2 presents the parts of individual accuracy scores for the selected machine learning algorithms. The *Y*-axis of each plot represents the number of models with different commissioning parameters that achieved the accuracy marked on the *X*-axis. The Random Forest Classifier was found to be the least sensitive to a variable number of input parameters. The Multi-Layer Perceptron is the least stable algorithm in our list. When designing a clinical decision support program, an appropriate selection of input parameters should be considered.

In Figure 3, we visualized the frequency of the accuracy score depending on the selection of input features. The number of models, with particular accuracy, is presented on the *Y*-axis. The accuracy is plotted on the *X*-axis as a number ranging from 0.0 to 1.0, e.g., value 0.5 means that the range of accuracy is greater than 0.45 but not greater than 0.5. Extending the set of analyzed variables may improve the performance of the models if applied to a larger group of patients or a population with different characteristics.

A comparison of the accuracy of classifiers, depending on the size of the input data, is included in Table 2. The ML methods showed different effectiveness in predicting the remission of IgAN. The reduction in the number of input parameters improved the maximum accuracy of most of the classifiers. This can be explained by the fact that the classifier found a relationship that did not lead to a practical conclusion.

### 3.3. A Different Approach to Neural Networks

Among all the models, there is a class of models with better results on all data divisions. These are models in which the first hidden layer has 1 to 50 neurons, but the second hidden layer only has 1 to 7 neurons. Then, for each split, there is a model with this structure, which is an ideal classifier and has an AUC of 1.0. We simulated a voting model that checks several others and chooses the best one or chooses the average of the individual models’ results. The average AUC for these models is 0.8842, and 0.9035 for the top ten average performance values. UPCR and sCr are key parameters related to the assessment of the possibility of remission. Combined with ALB and TP, it allows for effective prediction of possible remission. It is possible that for another group of patients there is a different contribution of features in prediction. UPCR reflects the loss of protein in the urine, while ALB and TP reflect the serum protein profile [13,14]. UPCR has the greatest validity of all, on average 20.71%, for the model selecting the relevant parameters, from an initial set of 35 features, the others are TP, ALB, TCh, and sCr with a validity of 7.18%, 6.71%, 5.20%, and 3.38%, respectively. The models built on these variables allowed the construction of models with different performances. It is possible that for a larger patient database, other measured input variables, the importance of the parameters will have a different distribution.

### 3.4. Regression Model

We have prepared a test for selected regression methods to select models with the best fitness and the smallest error in predicting the difference in creatinine levels at the mean follow-up time of 828 days (3–2825 days, SD = 748 days). We measured the performance of regression using the coefficient of determination R^2^, mean absolute error (MAE), and rooted-mean-square error (RMSE). Regression models were compared in terms of goodness-of-fit to training data, as measured with the R^2^ determination coefficient. The key goal, however, was to measure the accuracy of prediction of renal function deterioration. We measured it using the MAE and RMSE. Support-Vector Regression (SVR) with RBF (radial basis function) Kernel showed an MAE value of 0.1845 and RMSE of 0.2613, but a poor fit to the training data. The input data included sCr and UCPR values. Decision Tree Regressor showed MAE and RMSE values of 0.2059 and 0.2645 with the following input data: IgM deposits, the extent of interstitial fibrosis, platelets count, platelets to lymphocytes ratio, serum uric acid concentration, fasting glucose, serum albumin, total protein, and UPCR. It also yielded an R^2^ value of 0.9928 for the training data.

## 4. Discussion

The small amount of available data limits the use of machine learning on a large scale, especially outside the home center. This does not mean that it is completely ineffective, but great care should be taken in broad-range use. However, the flexibility of artificial intelligence allows the model to be trained through batching of data. Successively, the model achieves better performance as it overwrites the existing model. Our best ANN class had layers of up to 50 and up to 7 neurons. Compared to Schena et al., it is a model over 44 times smaller, and therefore with a lower risk of overfitting [15]. Despite the perfect fit to the training data, the models achieved very high accuracy on the test data. In the future, the system should be allowed to identify important data through a preliminary model. In our case, such a model was much less effective, but it allowed us to select the most important features that were grouped into subsets for the relevant models. Chen et al. designated a subset of 10 features from the initial total number of 36 variables [16]. Compared to the work of Han et al. we extended the functionality of the program with the possibility of assessing the rate of decline in renal function, expressed as a change in creatinine concentration [17].

In our work, we have demonstrated the influence of both model selection and parameters on the final results, and possible practical applications. Clear distortions appeared for the multi-parameter neural network input which was eliminated by reducing the number of variables. The Random Forest Classifier is a certain benchmark in measuring the accuracy of various designs, due to its stability and good performance, in relation to the rapid creation of models. Therefore, for some issues, it may be preferable to apply the random forest classifier rather than a neural network. It should be emphasized that the effectiveness of models extended with ALB, TP, or TCh is not inferior and may be used in practice. By such a gradual analysis of small subsets, we presented the crystallization of the optimal solution. It is the greed of the algorithm to choose the best models and evaluate them. The crucial influence of UPCR is based on its essential impact on the pathogenesis and progression of the disease. It is inherent to other input data sets. The number of possible combinations of inputs is extremely large, but we have only chosen those with significantly higher significance in terms of prediction. We have emphasized that with the growing number of the designs analyzed, the performance of the models changed. Therefore, we focused on a flexible solution, taking into account slightly weaker models built on three, four, or even more parameters.

The great advantage of a neural network is the possibility of expanding and estimating human perception. We have demonstrated that we should not speak of one effective model, but rather of a group of effective models, that is, a class of solutions to the classification problem. Then the full potential of both ANNs and modern computers is exploited. The preparation of a regression model with satisfactory goodness-of-fit was not an obstacle. The real challenge was to find a regressor with adequate performance, as measured by MAE and RMSE. Depending on the further application, it might be a search for the best fit, the lowest MAE and RMSE, or both. In our work, we have chosen the model that most effectively estimated disease progression and the compromise between fitting to data and minimizing errors. The smallest errors were achieved with Support-Vector Regression, with UPCR and sCR input parameters, but at the cost of lower goodness-of-fit to the training data. The Decision Tree Regressor turned out to be a compromise model.

At the same time, the abovementioned authors showed that the Random Forest Classifier had the best performance, which in our work was comparable to that of ANN. We have shown that the selection of a classifier should also be based on its performance depending on the variability of input data distribution. Our study was based on a small number of samples, but we found an idea to solve the problem by adopting an alternative approach, that is, searching for many optimal solutions and inferring from several outputs. When developing a clinical decision support system (CDSS), care should be taken with the choice of machine learning method. Random Forest Classifier, which is a technique simpler than ANN, showed the greatest stability depending on the choice of the input data. In the case of ANN, simple models can prove to be as effective as very complex ones. Our study has some strengths: we made a comparison of machine learning methods of varying complexity. In our work, we checked several ML techniques with which we built models of various structures. Thus, we found a group of models that achieved high efficiency regardless of the choice of input data. ML techniques are able to find important data for prediction. We have shown that the appropriate selection of the input data affects the efficiency of the model which is built on a small training set.

Machine learning is a powerful tool in medicine. It can significantly improve and support diagnostics. We have shown that the choice of input data and the choice of the right model have a direct impact on the performance. In order to build a CDSS, one should take the following into account: model; start parameters, if required by the method; and selection of input parameters. A model trained on one group of patients may be completely useless when applied to another group of patients in a different center. Therefore, we suggest that CDSS should include a module to find the best model. The limitation of our study is that it uses a retrospective design and a small number of samples. Our program can be expanded to include other ethnic groups due to flexibility, but more research is needed.

## Figures and Tables

**Figure 1 jpm-11-00312-f001:**
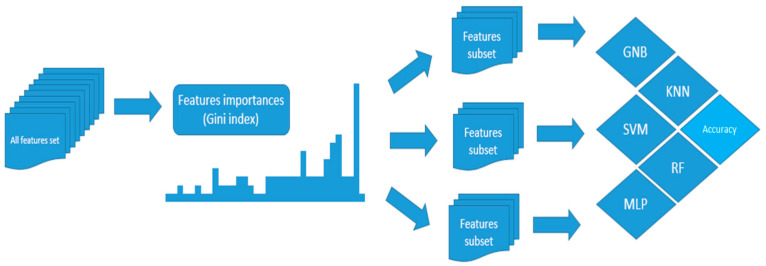
The Random Forest model is generated for the original database, then the parameters with the highest Gini index are selected and subsets are created from them. For these subsets, the final models are generated and evaluated, and analyzed for the best accuracy. GNB—Random Forest Classifier; KNN—K-nearest neighbor classifier; SVM—Support Vector Machines; MLP—Multi-Layer Perceptron; RF—Random Forest Classifier;

**Figure 2 jpm-11-00312-f002:**
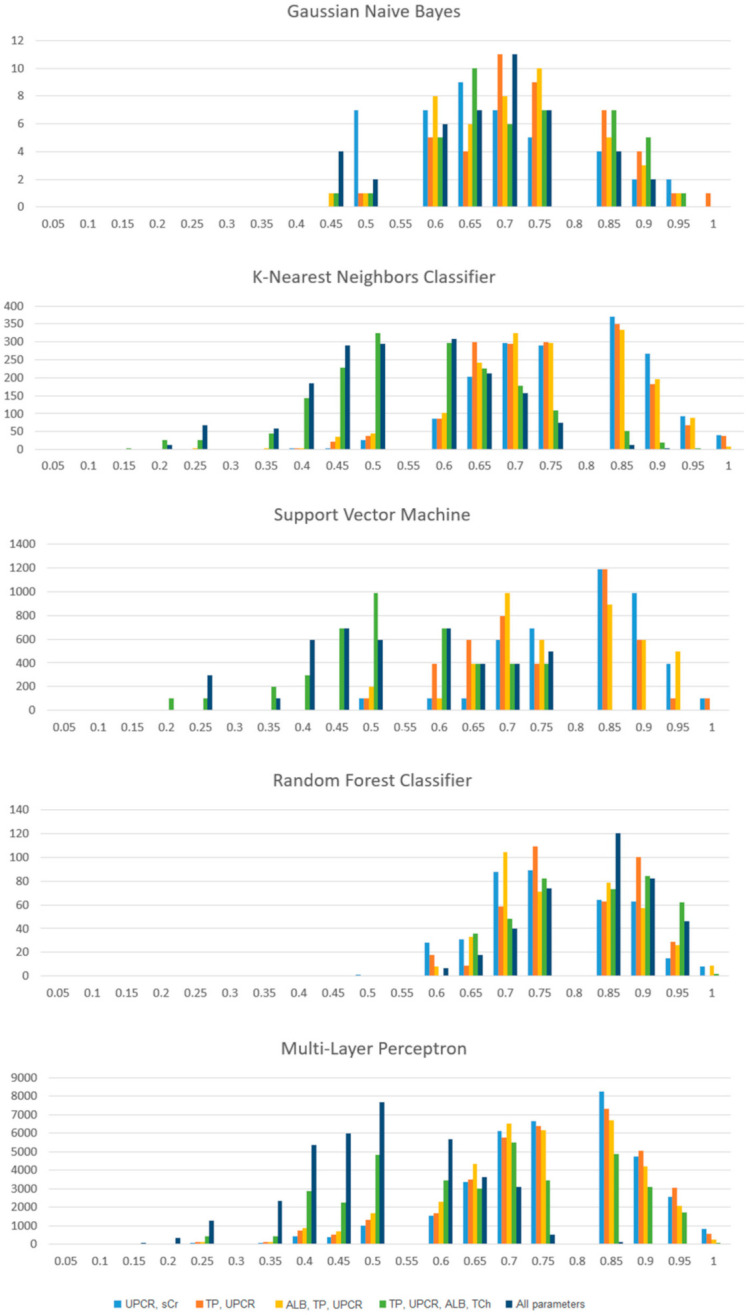
Different classifiers are characterized by different stability of accuracy depending on the selection of input features. Additional parameters can negatively affect classifier performance. Consider the type of classifier in practical application.

**Figure 3 jpm-11-00312-f003:**
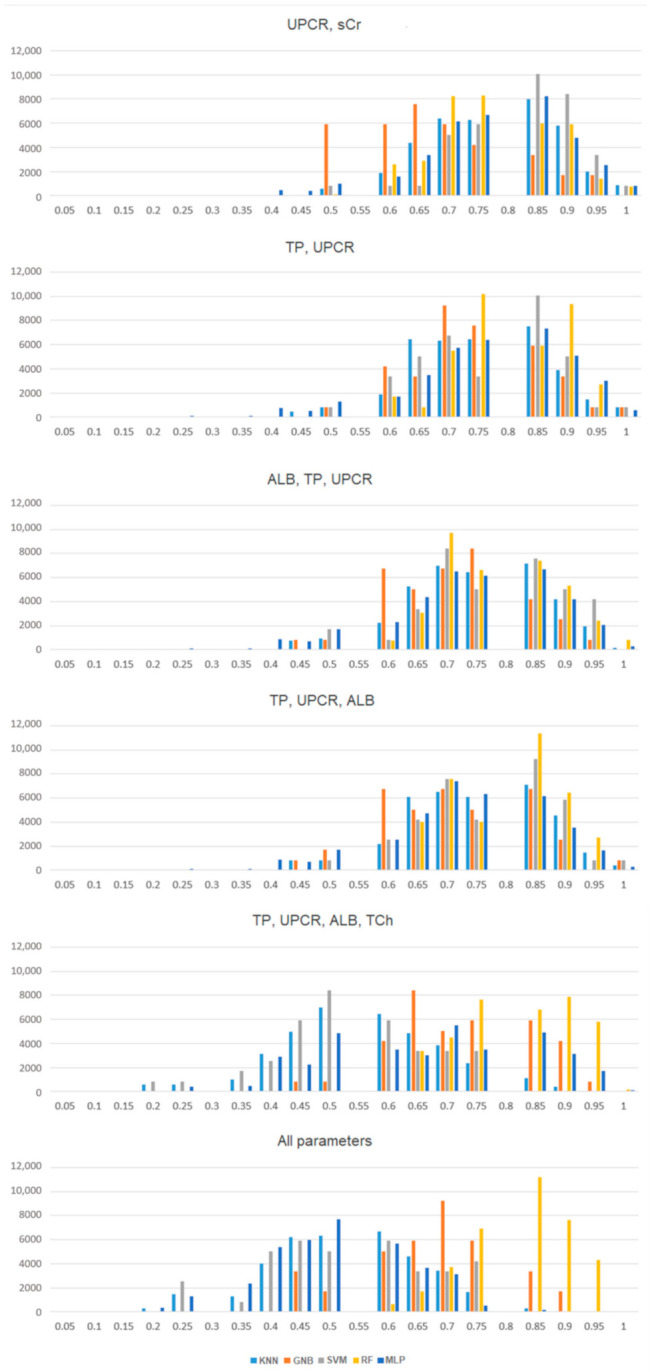
The classifiers have different sensitivities to the same input data. Consider the input data set in practical application. The histograms show the distribution of performance on a common scale.

**Table 1 jpm-11-00312-t001:** Population characteristics, assessed for 80 patients.

Feature	Values
	* *n* (%)
	Mean ± SD (Min-Max)
Female/Male (%)	35/35 (44%/66%) *
Age [years]	39.34 ± 13.89 (19–76)
EST-C score:	
M1	66 (82.5%) *
E1	24 (30.0%) *
S1	40 (50%) *
T2	1 (1.25%) *
C1	7 (8.75%) *
C2	1 (1.25%) *
Presence of IgM deposits	77 (96.25%) *
Interstitial fibrosis [%]	6.98 ± 8.13 (0–50)
SBP [mmHg]	131.25 ± 16.20 (100–170)
DBP [mmHg]	80.13 ± 11.19 (60–110)
MAP [mmHg]	95.46 ± 11.99 (72–128)
Erythrocyturia [RBC/HPF]	17.47 ± 15.51 (0.5–40)
WBC [ref. 4–10 10^3^/μL]	7.34 ± 2.43 (3.72–19.26)
NEU [ref. 2.5–6 10^3^/μL]	4.45 ± 2.00 (1.2–12.45)
LYM [ref. 1.5–3.5 10^3^/μL]	2.04 ± 0.77 (0.56–5.29)
PLT [ref. 140–440 10^3^/μL]	255.90 ± 64.34 (89–467)
NEU to LYM ratio (NLR)	2.58 ± 2.28 (0.65–19.16)
PLT to LYM ratio (PLR)	140.36 ± 58.62 (33.1–403.6)
Total cholesterol (TCh) [ref. 130–200 mg/dL]	258.38 ± 103.87 (121–753)
Triglycerides (TG) [ref. <150 mg/dL]	172.44 ± 91.42 (41–569)
Uric Acid [ref. 2.6–6 mg/dL]	6.57 ± 1.71 (3.2–11.1)
Fasting blood glucose [ref. 70–99 mg/dL]	92.79 ± 11.41 (69–141)
Serum albumin (ALB) [ref. 3.5–5.2 g/dL]	3.49 ± 0.92 (1.3–4.9)
Total protein (TP) [ref. 6.6–8.3 g/dL]	6.06 ± 1.20 (3.1–8.7)
Serum creatinine concentration (sCr) [ref. 0.7–1.1 mg/dL]	1.37 ± 1.36 (0.59–12.78)
eGFR [mL/min/1.73 m^2^]	68.40 ± 24.09 (5–135)
UPCR [g/g]	1.85 ± 2.17 (0.04–10.13)
Serum creatinine concentration difference	0.16 ± 2.27 (−11.74–15.32)
ACE inhibitors use	52 (65.0%) *

Abbreviations: white blood cells, WBC; lymphocytes, LYM; neutrophiles, NEU; platelets, PLT; systolic blood pressure, SBP; diastolic blood pressure, DBP; mean blood pressure, MAP; urine protein to creatinine ratio, UPCR; glomerular filtration rate, eGFR; standard deviation, SD; angiotensin-converting enzyme, ACE; laboratory reference value, ref. Data marked by * are number (n) and percentage of in contrary to the rest presented as mean, standard deviation and min and max.

**Table 2 jpm-11-00312-t002:** Comparison of the accuracy of classifiers depending on the size of the input data. The ML methods showed different effectiveness in predicting the remission of IgAN.

Algorithm	Input Parameters	AVG	SD	MIN	MAX
KNN	UPCR, sCr	0.7599	0.1109	0.3750	1.0000
	TP, UPCR	0.7367	0.1151	0.3750	1.0000
	ALB, TP, UPCR	0.7323	0.1172	0.2500	1.0000
	TP, UPCR, ALB	0.7317	0.1183	0.3125	1.0000
	TP, UPCR, ALB, TCh	0.5449	0.1368	0.1250	0.9375
	All parameters	0.5152	0.1279	0.1875	0.8750
GNB	UPCR, sCr	0.6628	0.1236	0.5000	0.9375
	TP, UPCR	0.7267	0.1100	0.5000	1.0000
	ALB, TP, UPCR	0.6933	0.1123	0.4375	0.9375
	TP, UPCR, ALB	0.6933	0.1226	0.4375	1.0000
	TP, UPCR, ALB, TCh	0.7064	0.1170	0.4375	0.9375
	All parameters	0.6584	0.1160	0.4375	0.8750
SVM	UPCR, sCr	0.7980	0.1016	0.5000	1.0000
	TP, UPCR	0.7427	0.1146	0.5000	1.0000
	ALB, TP, UPCR	0.7602	0.1143	0.5000	0.9375
	TP, UPCR, ALB	0.7500	0.1112	0.5000	1.0000
	TP, UPCR, ALB, TCh	0.5218	0.1334	0.1875	0.7500
	All parameters	0.5189	0.1424	0.2500	0.7500
RF	UPCR, sCr	0.7547	0.1034	0.5000	1.0000
	TP, UPCR	0.7854	0.0931	0.5625	0.9375
	ALB, TP, UPCR	0.7682	0.0986	0.5625	1.0000
	TP, UPCR, ALB	0.7791	0.0916	0.6250	0.9375
	TP, UPCR, ALB, TP	0.8009	0.0973	0.6250	1.0000
	All parameters	0.8025	0.0879	0.5625	0.9375
MLP	UPCR, sCr	0.7527	0.1261	0.2500	1.0000
	TP, UPCR	0.7450	0.1366	0.2500	1.0000
	ALB, TP, UPCR	0.7201	0.1372	0.1875	1.0000
	TP, UPCR, ALB	0.7104	0.1339	0.2500	1.0000
	TP, UPCR, ALB, TCh	0.6465	0.1712	0.1875	1.0000
	All parameters	0.4901	0.1230	0.0000	1.0000

Abbreviations: K-nearest neighbor classifier, Gaussian Naive Bayes, GNB; Support Vector Machines, SVM; Random Forest Classifier, RF; Multi-Layer Perceptron, MLP; urine protein to creatinine ratio, UPCR; serum creatinine concentration, sCr; serum albumin, ALB; total protein, TP; Total cholesterol, TCh.

## Data Availability

Data is contained within the article.

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
