# Peer review of "Machine Learning in Prediction of IgA Nephropathy Outcome: A Comparative Approach"

_jpm, 2021, doi:10.3390/jpm11040312_

Round 1

Reviewer 1 Report

Authors have presented the work well and showed how machine learning can be helpful in contributing to disease prediction. I have some minor comments,

1) Please mention the Y and X axis of the figures in the manuscript.

2) Elaborate the discussion with focus on the results you mentioned in the manuscript

3) There are some typos and grammar mistakes in the manuscripts, please proof read the manuscript before submission.

Author Response

Thank you for an effort you have made while reviewing our paper. We corrected the manuscript according to your suggestions.  

1) Please mention the Y and X axis of the figures in the manuscript.

The both Y and X-axis of the figures were mentioned in text. Lines 219-220.

2) Elaborate the discussion with focus on the results you mentioned in the manuscript

The paragraph has been added to the discussion. Lines 287-315

3) There are some typos and grammar mistakes in the manuscripts, please proof read the manuscript before submission.

We have corrected all mistakes in the manuscript. The paper underwent also extensive English editing, performed by certified office. 

Reviewer 2 Report

Konieczny et al. performed very interesting research regarding machine learning in prediction of IgA nephropathy outcome: a comparative approach. They found that application of machine learning methods requires careful selection of models and assessed parameters related to IgA nephropathy. I recommend the manuscript for the publication in JPM. 

Author Response

Thank you for the effort you have made in reviewing  our paper. We appreciate your remarks and positive comments.

Reviewer 3 Report

The manuscript describes an interesting approach to develop a diagnostic system to predict nephropathy using data from 2010 to 2019. I believe that the results obtained are useful and important.

Author Response

We appreciate your review and positive comments. Thank you for remarks and effort. 

Reviewer 4 Report

The authors examined by comparing different regression techniques, remission of proteinuria and the deterioration of kidney function, in patients with IgA nephropathy. The study was a very interesting result, but some considerations have been left behind.

The author has carefully describes the incorporation of the analysis subject. To better understand, it is recommended that shown by the flow of the subject selection.

Table 1.:

It adds the normal range (reference value) for the background factor of the analysis subject. Range of "Valles" in the table is the standard error (SE)? It adds the number of analysis subject.

The author was prepared using the Python programming language and the available scikit-learn library a program, creating 5 machine learning models. Combined with ALB and TP, it was decided to allow the effective prediction of the possible remission. Is there not a promising combination with other factors? Add to considered in conjunction with the features of the disease.

Author Response

  1. The author has carefully describes the incorporation of the analysis subject. To better understand, it is recommended that shown by the flow of the subject selection.

    Regarding to patients in analyzed group, we included all subjects with IgA nephropathy, for whom complete clinical and histological data were available.

    Final decision which factors would be included into analysis, depended on analytic algorithm, without excluding any data.

  2.  

    Table 1.:

    It adds the normal range (reference value) for the background factor of the analysis subject. Range of "Valles" in the table is the standard error (SE)? It adds the number of analysis subject.

    We have corrected Table 1, by adding reference values for laboratory parameters.  Moreover, we added explanations allowing understanding of presented data. Numerical parameters were presented as mean, standard deviation and minimum and maximum.
  3. The author was prepared using the Python programming language and the available scikit-learn library a program, creating 5 machine learning models. Combined with ALB and TP, it was decided to allow the effective prediction of the possible remission. Is there not a promising combination with other factors? Add to considered in conjunction with the features of the disease.

    UPCR has the greatest validity of all, on average 20.71%, for the model selecting the relevant parameters, from initial set of 35 features, the others are TP, ALB, TCh and sCr with 7.18%, 6.71%, 5.20% and 3.38%, respectively. The models built on these variables allowed construction of models with different performance. It is possible that for a larger patient base, other measured input variables, the importance of the parameters will have a different distribution.